# Evaluation of Carboxymethyl Cellulose/Gelatin Hydrogel-Based Dressing Containing Cefdinir for Wound Healing Promotion in Animal Model

**DOI:** 10.3390/gels11010038

**Published:** 2025-01-04

**Authors:** Zahra Soleimani, Hadi Baharifar, Najmeh Najmoddin, Kamyar Khoshnevisan

**Affiliations:** 1Department of Biomedical Engineering, Science and Research Branch, Islamic Azad University, Tehran 1477893855, Iran; 2Applied Biophotonics Research Center, Science and Research Branch, Islamic Azad University, Tehran 1477893855, Iran; 3Medical Nanotechnology and Tissue Engineering Research Center, Shahid Beheshti University of Medical Sciences, Tehran 1983963113, Iran; 4Department of Tissue Engineering and Applied Cell Sciences, School of Advanced Technologies in Medicine, Shahid Beheshti University of Medical Sciences, Tehran 1983963113, Iran; 5Research and Development Team, Evolution Wound Dressing (EWD) Startup Co., Tehran 1983963113, Iran

**Keywords:** wound healing, hydrogel, carboxymethyl cellulose, gelatin, cefdinir

## Abstract

The skin serves as a critical barrier against external pathogens, and its wound healing is a complex biological process that requires careful management to ensure optimal tissue regeneration. Hydrogels, a class of hydrophilic polymers, have emerged as promising materials for wound dressings due to their biocompatibility, biodegradability, and ability to create a moist wound environment conducive to cell proliferation and migration. In this research, a hydrogel dressing containing cefdinir (Cef) was made from a combination of carboxymethyl cellulose (CMC) and gelatin (Gel) by a physical crosslinking method, and their physicochemical, mechanical, and biological properties were investigated. Results show that the addition of Cef does not cause a significant change in the morphology or the tensile strength of the wound dressing. The swelling and degradation rate of the hydrogel slightly increased in the presence of Cef. The presence of Cef enhanced antibacterial effects up to 2.5-fold against *P. aeruginosa* (35 mm), *S. aureus* (36 mm), and *S. pyogenes* (35 mm). The results of the cytotoxicity test showed the absence of cytotoxicity in both drug-containing and drug-free wound dressings, as well as a survival rate of over 75% in cells after 48 h. The drug-containing wound dressing accelerates the formation of the epidermis layer and the production of fibroblast cells, and as a result, accelerates the wound healing process. The percentage of wound healing on the ninth day of treatment for an untreated wound was 30%, while this percentage was 40% with a wound dressing without medicine and 60% with a wound dressing containing medicine, and on the fifteenth day of treatment, the wound treated with both wound dressings had more than 85% healing. As a result, it is possible to use CMC/Gel hydrogel polymeric wound dressing containing Cef as a wound dressing for wound healing, according to the desired physicochemical properties and biocompatibility.

## 1. Introduction

The skin is the body’s outermost layer and comes into close contact with the environment. Thus, injuries, illnesses, and burns can alter its composition and function, disrupting the tissues in the afflicted area [1,2]. Acute wounds can heal spontaneously after around 8 to 12 weeks. However, chronic wounds need more attention and may even require surgery. For these reasons, wound care in the global community is confronted with a multitude of issues, such as the rising incidence of type 2 diabetes, obesity, aging populations, and the demand for reasonably priced wound dressings [3,4]. The ideal wound dressing has the following characteristics: maintaining high humidity, removing excess wound secretions, antibacterial, non-toxic, favorable degradability, non-sticky, comfortable, dry, and high biocompatibility [5,6]. The variability in wound characteristics necessitates a personalized approach to wound dressing selection to optimize healing. Dressings range from traditional options suitable for clean, dry wounds with minimal exudate, to modern moisture-retaining varieties. Layered contact dressings, and thin non-adherent sheets, protect the wound bed from direct contact with other materials. Modern options include semi-permeable films, foams, hydrocolloids, hydraulic fibers, superabsorbent dressings, composites, pharmaceutical dressings, sprayable dressings, and hydrogels [7,8].

Traditional methods, such as bandages and gauze, are effective in controlling bleeding. However, they suffer from several limitations, including non-biodegradability, susceptibility to infection, secondary tissue damage, and limited effectiveness in wound healing [9]. Wound dressings are primarily produced from biological and synthetic polymers. In recent decades, a growing trend has emerged towards advanced wound dressings, such as natural/synthetic polymer dressings, polymer film dressings, hydrocolloid dressings, and hydrogel dressings [10,11]. Hydrogels are favored due to their biocompatibility, swelling properties, and mechanical similarity to extracellular matrices. They also excel in secretion absorption, oxygen transfer, and deformation, which are crucial for wound healing [12]. Moreover, hydrogels naturally contain a large amount of water, aiding in wound hydration [13,14]. Cross-linking, a stabilization process in polymer chemistry, enhances the strength, extensibility, thermal stability, and repairability of polymers, making them suitable for various applications [15,16]. The selection of appropriate materials is crucial for fabricating effective hydrogel dressings. Carboxymethylcellulose (CMC), a natural polymer, is a promising candidate due to its biocompatibility, biodegradability, tissue simulation, low cost, and non-toxicity [17,18].

Gelatin (Gel) is a valuable co-product of the meat industry used to create gel-based hydrogels. In these hydrogels, Gel forms the cross-linked polymer network, providing the characteristic gel properties [19,20]. Gel-based hydrogels have attracted significant attention as bioactive matrices due to their biocompatibility, biodegradability, and ability to undergo chemical modification for various functionalities [21]. Bacterial infection and wound colonization significantly increase healthcare costs by delaying recovery [22]. Healthy skin harbors bacteria that play a vital role in its ecosystem. When the skin barrier is broken, these bacteria can migrate from the surface to areas where they are not normally present, causing an imbalance that leads to infection in the injured area [23].

One approach to treating wound infections involves bandages with antibacterial chemicals on their surface that can slowly release into the damaged region [24]. Effective antibacterial wound dressings have gained considerable research interest due to their ability to inhibit the growth of both Gram-positive and Gram-negative bacteria in the wound area [25].

Cefdinir (Cef) exhibits a broad spectrum of antibacterial activity, commonly associated with respiratory and skin infections [26]. Novel wound care treatment options are urgently needed to address the increasing number of severe, acute, and chronic wounds in today’s aging population. Although numerous hydrogel-based dressings are currently on the market, they often suffer from high costs, safety concerns, and limitations in mechanical stability. To address these limitations and accelerate wound healing, we aimed to develop a CMC-based hydrogel loaded with Cef to maintain a moist wound environment and minimize healing time. The developed hydrogel-based dressing was subjected to various tests, including cytotoxicity, antibacterial activity, and histopathological analysis in an animal model.

## 2. Results and Discussion

### 2.1. Characterization of Hydrogels

The hydrogel films exhibited a transparent, smooth surface with an average thickness of 200–300 μm post-drying. SEM micrographs (Figure 1) revealed a porous structure, with pore size and density increasing significantly in the cefdinir-incorporated hydrogel (Figure 1b1,b2). The CMC/Gel hydrogel demonstrated a porosity of approximately 27%, while the cefdinir-containing hydrogel exhibited a porosity of 43%. The presence of cefdinir likely disrupted the collagen matrix, reducing inter-polymer connections and promoting pore formation.

The porous structure of the hydrogel offers several benefits for wound healing applications, including enhanced drainage, prevention of exudate accumulation, and the creation of an optimal environment for antibiotic diffusion [27]. A high porosity, particularly with pore sizes in the 20–100 μm range, is ideal for facilitating cell infiltration and neovascularization [28,29]. While crosslinking generally reduces porosity, the incorporation of cefdinir appears to counteract this effect, resulting in a more porous structure that is beneficial for wound healing.

### 2.2. Fourier Transform Infrared Analysis

FTIR spectroscopy was employed to investigate the chemical composition and potential interactions between the components of the hydrogels. The FTIR spectra of the hydrogels are depicted in Figure 2. The spectrum of the CMC/Gel hydrogel (Figure 2a) exhibits characteristic peaks associated with the functional groups present in the constituent polymers. The broad peak centered around 3593 cm^−1^ corresponds to the O–H stretching vibration of the hydroxyl groups in CMC. The peak at 1756 cm^−1^ is attributed to the C=O stretching vibration of the ester groups formed during the crosslinking reaction between CMC and Gel. The bands at 1451 cm^−1^ and 1530 cm^−1^ are assigned to the amide I and amide II bands of the peptide linkages in Gel, respectively. The peak at 1451 cm^−1^ is also indicative of the interaction between the anionic groups of CMC and the cationic groups of Gel [30].

The FTIR spectrum of the CMC/Gel/Cef hydrogel (Figure 2b) shows additional peaks compared to the CMC/Gel hydrogel. The peaks at 1426 cm^−1^ and 1252 cm^−1^ are characteristic of the oxime and C–N stretching vibrations of cefdinir, respectively, confirming its successful incorporation into the hydrogel matrix [31].

### 2.3. Mechanical Measurement

The mechanical properties of a wound dressing are crucial for optimal performance. It should exhibit sufficient flexibility to conform to the contours of the wound site and withstand mechanical stress during movement. Figure 3 illustrates the tensile stress–strain curves of the CMC/Gel and CMC/Gel/Cef hydrogels.

The CMC/Gel hydrogel demonstrated higher tensile strength and Young’s modulus (25.4 kPa) compared to the CMC/Gel/Cef hydrogel (18.6 kPa). This reduction in mechanical properties in the Cef-containing hydrogel can be attributed to the formation of chemical bonds between the drug molecules and the polymer chains [32]. While these interactions enhance the drug release profile and antimicrobial properties, they may compromise the overall mechanical integrity of the hydrogel. However, the reduced stiffness of the CMC/Gel/Cef hydrogel may be beneficial in certain wound healing scenarios, as it can improve conformability and reduce irritation.

### 2.4. Swelling Behavior

One of the practical features of a hydrogel is its high water absorption capacity [6]. The swelling property of wound dressings ensures the absorption of exudate and maintains a moist wound environment [33]. Given that the pH of acute wounds is typically around 7.44, while chronic wounds exhibit a pH range of 7.42 to 8.90 [34], the swelling, degradation, and drug release behavior of the hydrogel was evaluated at pH 7.4 and 9.0. Figure 4 illustrates the percentage weight change in the hydrogels over 48 h at different pH levels. Both samples displayed an initial rapid swelling phase across all pH conditions, driven by their inherent porosity and crosslinking structure. The incorporation of Cef enhanced pore size and porosity, leading to a significant increase in the swelling rate from 8 to 16 h. The maximum swelling ratio for the CMC/Gel hydrogel in neutral pH was approximately 76%, achieved after 20 h, while the addition of Cef elevated this value to around 86% within 16 h.

As the pH increased, both samples exhibited a corresponding increase in swelling rate and extent. At alkaline pH, the sample containing Cef demonstrated a more pronounced swelling increase, reaching a final value of 93% at 16 h, compared to 82% for the drug-free sample. The highest swelling rate was observed for the Cef-loaded hydrogel at pH 9 (Figure 4), while the lowest rate was associated with the drug-free hydrogel at neutral pH. The increased swelling rate and extent at alkaline pH can be attributed to the deprotonation of carboxyl groups in gelatin and CMC, resulting in enhanced negative charge and electrostatic repulsion, as reported in previous studies [35,36]. Both samples, in both neutral and alkaline pH conditions, have the potential to provide adequate moisture to the wound during the healing process [37].

As illustrated in Figure 4, a positive correlation was observed between swelling rate and degradation rate. The enhanced pore size and porosity induced by Cef [37], coupled with the generation of negatively charged groups at alkaline pH, primarily contributed to the accelerated degradation [35]. Consequently, the degradation rate at pH 9 was generally higher than that at pH 7.4. The hydrogel incorporating cefdinir at alkaline pH exhibited the most pronounced degradation, reaching 81% within 48 h. Under identical conditions, the drug-free sample demonstrated 67% degradation. In all samples, the onset of degradation initiated approximately 20 h after water absorption, except for the drug-free CMC/Gel sample at neutral pH, where degradation initiated at 24 h and proceeded at a significantly slower rate. This sample ultimately achieved a 57% degradation rate at 48 h, indicating a lower degradation extent compared to the cefdinir-containing sample under similar conditions.

The rate of degradation plays a pivotal role in the repair and replacement of natural body structures with scaffolds and dressings. Ideally, the degradation rate should align with tissue regeneration rates. However, achieving this synchronization can be challenging in practice. In the context of dressings, while the precise degradation rate is less critical due to their ease of replacement, a slower degradation rate can enhance patient comfort by reducing the frequency of dressing changes [38,39].

### 2.5. Release Profile

Drug release from the Cef-containing sample was evaluated at both neutral and alkaline pH. As depicted in Figure 5, Cef release commenced concurrently with hydrogel swelling at both pH levels. The absence of strong drug–polymer interactions facilitated the release process, and the accelerated swelling and degradation rates further promoted its release. As anticipated, the release rate was significantly higher at alkaline pH, particularly at 12 to 20 h, compared to neutral pH.

The initial rapid release at both pH levels can be attributed to the concentration gradient between the hydrogel and the surrounding medium [40]. The cumulative drug release reached 83% after 24 h at alkaline pH and 79% at neutral pH. As the hydrogel degradation progressed, a slight increase in release was observed in both samples up to 48 h, potentially due to a reduction in the drug release rate resulting from the decreased initial drug concentration.

The rapid initial drug release could effectively address acute infections [41]. However, it is important to note that the in vivo release profile may be influenced by factors such as drug consumption and the interaction between the antibiotic and the wound environment.

### 2.6. Antibacterial Activity

The antibacterial effect of both hydrogels was investigated against *S. aureus*, *S. pyogenes*, and *P. aeruginosa*. The antibacterial test results are shown in Figure 6. According to the images of the plates, CMC/Gel hydrogel was not observed with a significant inhibition zone, while the CMC/Gel/Cef exerted a magnificent antimicrobial effect against the growth of all three bacterial strains.

The inhibitory zone of the CMC/Gel hydrogel against all three bacterial strains was approximately the same at about 14 ± 2. The CMC/Gel/Cef hydrogel exhibited the largest inhibition zone at about 36 ± 2 mm against *P. aeruginosa*. CMC/Gel/Cef antibacterial activity against *S. aureus* and *S. pyogenes* was almost equal at about 35 ± 2. Cef was the most active drug among cephalosporins against oxacillin-sensitive *S. aureus* and coagulase-negative staphylococci, S. pneumoniae, *S. pyogenes*, E. coli, and M. catarrhalis [42]. Cef’s antibacterial activity is affected by the binding of penicillin to proteins [31]. Very few studies have investigated the antibacterial inhibitory activity of Cef on *P. aeruginosa*. Based on the results obtained from the disk diffusion technique in our research, it is clear that the dressing with CMC/Gel/Cef can successfully inhibit bacterial activity in wounds.

### 2.7. MTT Assay Results

The cytotoxicity of the CMC/Gel and CMC/Gel/Cef hydrogels was assessed and is presented in Figure 7. After 24, 48, and 72 h of incubation, the cell survival percentage for the CMC/Gel hydrogel was consistently higher than that of the Cef-containing hydrogel. Notably, the cell viability of the CMC/Gel/Cef hydrogel decreased significantly after 72 h compared to the CMC/Gel hydrogel.

Antibacterial agents can indeed exhibit toxicity toward eukaryotic cells when they accumulate in cell culture media. However, this phenomenon is unlikely to significantly impact cell survival in vivo due to several factors. Firstly, the use of Cef, a specific antibacterial agent, targets bacterial cells and minimizes harm to host cells. Secondly, the presence of exudate, a physiological fluid that can dilute the concentration of the antibacterial agent, may help prevent excessive accumulation around the pores of the hydrogel.

### 2.8. Animal Study

The healing process of standard wounds in different animal groups is illustrated in Figure 8. In this study, the animal models exhibited satisfactory recovery post-surgery, with no signs of infection. Throughout the perioperative period, no surgical complications arose that could potentially compromise the test results. Furthermore, the animals maintained a healthy condition postoperatively.

After five days, the control group’s wounds displayed a dried appearance, whereas the hydrogel-treated groups exhibited a moist environment between the hydrogel membrane and the wound bed. By day nine, the treated groups demonstrated a higher abundance of granulated tissue with a more pronounced red coloration compared to the control group. In terms of wound size, the treated wounds exhibited a more rapid reduction rate than the untreated wounds, with a noticeable difference between the CMC/Gel- and CMC/Gel/Cef-treated groups on the fifteenth day of the experiment.

The wound closure rate is depicted in Figure 9. The figure reveals that on the initial day, there was no significant difference in wound closure rates among the three animal groups. However, from the third day onward, the hydrogel-treated groups exhibited accelerated healing compared to the control group. On this day, the wound closure rate in the hydrogel-treated groups reached approximately 20%, significantly surpassing the control group. This accelerated healing can be attributed to the moist wound environment created by the hydrogel.

Between the fifth and eleventh days of the study, the group receiving the Cef-containing hydrogel demonstrated significantly superior healing compared to the group receiving the hydrogel alone. On the seventh day, the wound closure rate in the Cef-containing hydrogel group increased to 60%. It is likely that the antimicrobial action of Cef during this period contributed to reduced acute infections and subsequent accelerated healing. Throughout days 5 to 11, both hydrogel-treated groups exhibited significantly higher wound healing rates compared to the control group. After the eleventh day, the rate of recovery in the hydrogel-treated groups became comparable, while still maintaining a significant advantage over the control group.

An ideal wound dressing material typically possesses several key characteristics: (1) biocompatibility with cellular tissues, whether derived from natural or synthetic sources; (2) minimal potential for inducing inflammation, disease transmission, or adverse host immune responses; and (3) an optimal microarchitecture that facilitates cell infiltration and efficient binding [43,44]. The hydrogel incorporating Cef appears to have the potential to accelerate wound healing. However, since the wounds were not infected, the effect of antibiotics on accelerating wound healing cannot be attributed to infection control. It appears that, in the present study, the presence of cefdinir in the hydrogel enhances exudate absorption by influencing hydrogel swelling. Additionally, it may be effective in eliminating bacteria introduced to the wound during dressing changes.

It is important to note that the widespread use of topical antibiotics raises concerns regarding the emergence of antimicrobial resistance. The susceptibility of the infecting bacteria to the specific antibiotic employed is a crucial factor in effective wound treatment. In this study, both drug-free and drug-loaded hydrogel formulations were evaluated in animal models. As depicted in Figure 9, the base hydrogel demonstrated the potential to accelerate wound healing by influencing coagulation, maintaining a moist environment, and absorbing exudate. Therefore, antibiotic-containing hydrogels may prove beneficial in cases of potential wound infection, particularly when considering the specific bacterial etiology.

Histopathological images and detailed data are presented in Figure 10 and Table 1. According to the tabulated results, the number of epidermal layers in wounds treated with both hydrogels at day 9 exceeded that of untreated wounds (score of 2 out of 4). Furthermore, at day 15, the number of epidermal layers in wounds treated with drug-containing dressings increased.

According to the edematous process, acute-phase inflammatory cells, primarily neutrophils, infiltrate the wound site in the initial days following injury. These cells are typically more prominent during the first five days. Subsequently, the chronic phase commences, marked by the replacement of neutrophils with mononuclear inflammatory cells. Between the seventh and tenth day post-skin lesion, a decrease in inflammatory cells is observed, and by the fourteenth to twenty-first day, they usually diminish significantly or disappear entirely [45]. Therefore, the presence of neutrophils for up to three days is considered normal. Typically, neutrophils decline after the fifth day, and their place is taken by lymphocytes and macrophages. By the tenth day, the overall population of inflammatory cells decreases. The results obtained from this study demonstrated a reduction in the number of inflammatory cells in both hydrogel-treated wounds on the fifteenth day, whereas they were still present in the untreated wound (Table 1).

The formation of new blood vessels in the early stages of wound healing indicates an active repair process. This angiogenic response typically diminishes in the second week as the wound progresses toward closure. According to the data presented in Table 1, the decline in angiogenesis during the second week is more pronounced in hydrogel-treated areas, suggesting accelerated wound healing [45,46].

Fibroblast proliferation is closely linked to the interaction between lymphocytes and macrophages. The population of fibroblasts increases significantly from the second to the fourth day post-wounding. This proliferation continues until the fourteenth day, ultimately contributing to wound bed formation through the contraction of the fibrocytic–fibroblastic network [45]. A positive correlation exists between the extent of fibroblast proliferation and the rate of wound healing. In this study, the number of fibroblasts at day five was higher in wounds treated with the drug-containing dressing compared to the other two groups. Additionally, on the fifteenth day, both hydrogel-treated wounds exhibited a higher number of fibroblasts than the untreated wounds.

The CMC/Gel and CMC/Gel/Cef hydrogels effectively facilitated the transformation of fibroblasts into fibrocytes, as evidenced by the increased collagenogenesis observed in these groups compared to the control. Wound dressings can accelerate the healing process by influencing various aspects of wound repair. Key factors contributing to the efficacy of the CMC/Gel/Cef scaffold include the provision of a moist environment, absorption of exudate, antibacterial properties, and a suitable scaffold for fibroblast migration and proliferation.

## 3. Conclusions

In this study, CMC/Gel and CMC/Gel/Cef hydrogels were successfully synthesized and evaluated as potential wound dressing materials. We assessed their characteristics, including antibacterial activity, cell viability, and in vivo wound healing properties, supported by histopathological analysis. Surface characterization revealed a smooth and uniform morphology. The Cef-containing hydrogel exhibited superior swelling and degradation properties. Additionally, both hydrogels demonstrated adequate mechanical strength. Notably, both hydrogels exhibited significant antibacterial activity against *S. pyogenes*, *S. aureus*, and *P. aeruginosa*. Furthermore, both drug-free and drug-loaded hydrogels were found to be non-toxic. In vivo studies in rat models demonstrated that wounds treated with both hydrogels exhibited accelerated wound healing within 15 days. Histopathological analysis revealed that the wound dressings significantly promoted the formation of the epidermal layer and fibroblast cells, thereby enhancing the overall wound healing process. Overall, the CMC/Gel/Cef wound dressing holds significant potential for clinical application in wound healing.

## 4. Materials and Methods

### 4.1. Materials

CMC (Mw = 250 kDa, viscosity 735 cps, purity 99.5%), citric acid, and Cef were purchased from Sigma-Aldrich, St. Louis, MO, USA. Gel was purchased from Merck Millipore, Darmstadt, Germany. Other materials were attained from ascribed centers unless otherwise specified.

### 4.2. Synthesis of CMC/Gel and CMC/Gel/Cef Hydrogels

CMC/Gel hydrogel solution was prepared by adding carboxymethyl cellulose powder (2.0 g) and Gel powder (0.5 g) to 100 mL of DI water and stirring at 200 rpm at room temperature until complete solubilization occurred. The crosslinking agent, citric acid (5.0% *w*/*v*), was added and stirred at 400 rpm for 30 min to ensure homogenization. Then, 10 mL of the solution was poured into plastic molds (polystyrene Petri dishes with a diameter of 35 mm) and dried for 48 h at room temperature (21–25 °C) to remove water.

To prepare the CMC/Gel/Cef hydrogel, the same method was used, with the addition of Cef powder (0.01 g) to 5 mL of distilled water and stirring at 700 rpm for 1 h. The Cef solution (0.2 *w*/*v*) was slowly added to the hydrogel and stirred for 30 min at 600 rpm. Then, 10 mL of the solution was poured into plastic molds and dried for 48 h at room temperature to remove water [35,47].

### 4.3. Characterization of Hydrogels

#### 4.3.1. Morphology Observation

The surface and cross-sectional morphology of the hydrogel films were examined using scanning electron microscopy (SEM, AIS2100, Seron Technology, Anseong, Republic of Korea). Prior to imaging, the samples were sputter-coated with a layer of gold and examined under the microscope at 25 kV.

#### 4.3.2. Fourier Transform Infrared (FTIR) Spectroscopy

FTIR analysis was performed using a Thermo Nicolet AVATAR 360 FTIR spectrometer (Waltham, MA, USA). Samples (1 mg) were mixed with KBr powder (300 mg) under vacuum and analyzed using the KBr pellet method.

#### 4.3.3. Tensile Testing

Tensile strength measurements were performed using a universal testing machine (SANTAM, Model STM1, Karaj, Iran) equipped with a 1 kN load cell. Dried film specimens were cut into rectangular shapes measuring 3 × 5 cm. Samples were then placed between the jaws of the machine and pulled apart at a constant speed of 1 mm/min until rupture. Stress–strain curves were plotted, and data from triplicate measurements were averaged and expressed as the mean.

#### 4.3.4. Swelling and Degradation Evaluation

To measure water uptake capacity and degradation rate, hydrogel samples (1 cm^2^) were dried at room temperature (RT) to a constant weight (W_0_, initial weight) and then immersed in 20 mL of deionized water at pH 7.4 and 9.0. At specific time intervals, the hydrogels were removed, blotted dry on filter paper, and weighed again (W_1_). The percentage weight change was calculated in triplicate using Equation (1).
(1)$$weight\;change\%=(W_{1}-W_{0})/W_{0}\times 100$$

#### 4.3.5. Release Profile

In vitro release studies were conducted using a dialysis bag (MWCO 12 kDa). Five milliliters of the Cef-loaded hydrogel was enclosed within the dialysis bag and immersed in 50 mL of phosphate-buffered saline (PBS) at different pH levels (7.4 and 9.0). At predetermined time intervals, 1 mL of the release medium was withdrawn, and its absorbance was measured at 287 nm using a spectrophotometer (Cary 100, Agilent). The withdrawn volume was immediately replaced with fresh PBS, and the experiment was replicated three times at each time point [40].

#### 4.3.6. Antibacterial Activity

The antimicrobial activity of the hydrogels was determined against *S. aureus* (ATCC 25923), *S. pyogenes* (ATCC 19615), and *P. aeruginosa* (ATCC 27853). The disk diffusion method was used to evaluate the antimicrobial activity [48]. Bacterial suspensions, equivalent to half McFarland’s turbidity, were cultured on Mueller–Hinton agar. Disks of CMC/Gel and CMC/Gel/Cef, with a diameter of 1 cm, were placed on the agar plates and incubated for 24 h at 37 °C. Subsequently, the inhibition zone diameter of each sample was measured. Each cycle was repeated three times and the average value was calculated.

#### 4.3.7. Cytotoxicity Test

MTT assay was carried out to investigate the effect of CMC/Gel and CMC/Gel/Cef hydrogel dressing extracts on human foreskin fibroblast (HFF) cells. Normal HFF cells were obtained from the Pasteur Institute cell bank of Iran and cultured in DMEM containing 10% FBS and incubated at 5% CO2. Extracts from CMC/Gel and CMC/Gel/Cef hydrogel dressings were prepared. Then, 100 μL of each extract was added to 105 cells in each well of a 96-well plate and incubated for 72 h. At specific time points, the culture medium was removed, and 20 μL of MTT solution (Atocel, Budapest, Hungary) was added to each well and incubated for 4 h. Then, the supernatant was drained, and 100 μL of DMSO was added to each well. The absorbance at a wavelength of 590 nm was subsequently measured using a microplate reader (DA-3200, Garni, Rizpardaz, Tehran, Iran), and the cell viability percentage was calculated using Equation (2) [48].
(2)Viability%=(As−Ab/ Ac−Ab)×100
where As denoted the sample absorbance, Ab denoted the blank absorbance, and Ac denoted the control absorbance.

#### 4.3.8. In Vivo Investigation

Fifteen female white mice, weighing between 21 and 26 g, were used for the animal study. The animals were maintained at room temperature (23 °C) on a 12 h light/dark cycle with free access to standard rodent chow and water. The animal experiments were conducted following the guidelines established by the Ethics Committee for Animal Experiments at Science and Research Branch, Islamic Azad University (Ethical code number: IR.IAU.SRB.REC.1403.024). The studies also adhered to European regulations on the care and use of animals in experimental procedures and the ARRIVE guidelines. The mice were divided into three groups: Group A was treated with CMC/Gel, Group B was treated with CMC/Gel/Cef, and Group C served as the control group. Animals were anesthetized with xylazine–ketamine, and the hair on their backs was shaved. A 6 mm punch biopsy was used to create circular wounds on the backs of the animals. The day of wounding was designated as day zero. Dressings were changed daily. Images of the wounds were captured daily, and the wound area was measured. The wound closure percentage was calculated using Equation (3) [49].
(3)$$Wound\;closure\%=A_{0}-A_{t}/A_{0}\times 100$$
where A_0_ is the initial wound area, and A_t_ is the wound area at the time of the study.

Hematoxylin–eosin staining was employed to examine the healing process by examining epithelialization, inflammatory cell infiltration, fibroblast density, angiogenesis, and collagen deposition [45].

#### 4.3.9. Statistical ANALYSIS

Statistical analyses were performed using the SPSS software, version 20.0 (SPSS, Inc., Chicago, IL, USA). One-way ANOVA tests were employed to compare groups and *p* values ≤ 0.05 were considered to be significant.

## Figures and Tables

**Figure 1 gels-11-00038-f001:**
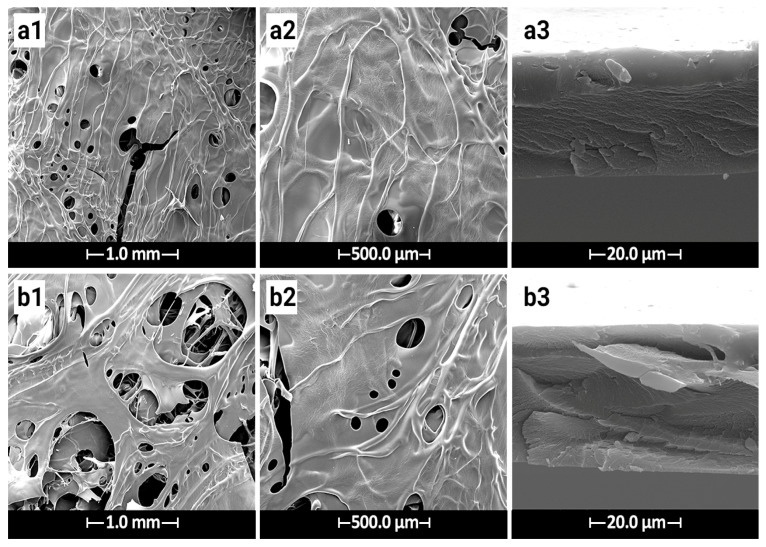
Scanning electron micrographs of hydrogel samples imaged at 20 kV: (**a1**,**a2**) surface morphology of CMC/Gel, (**a3**) cross-sectional morphology of CMC/Gel, (**b1**,**b2**) surface morphology of CMC/Gel/Cef, (**b3**) cross-sectional morphology of CMC/Gel/Cef.

**Figure 2 gels-11-00038-f002:**
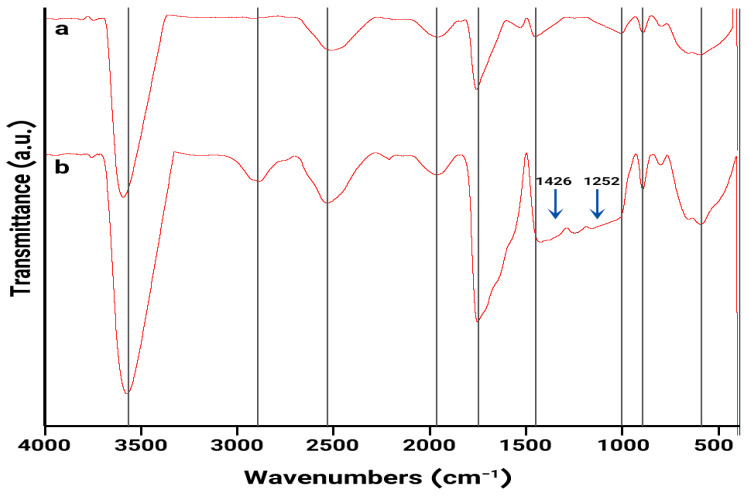
FTIR spectra of the CMC/Gel (**a**) and CMC/Gel/Cef (**b**) hydrogels.

**Figure 3 gels-11-00038-f003:**
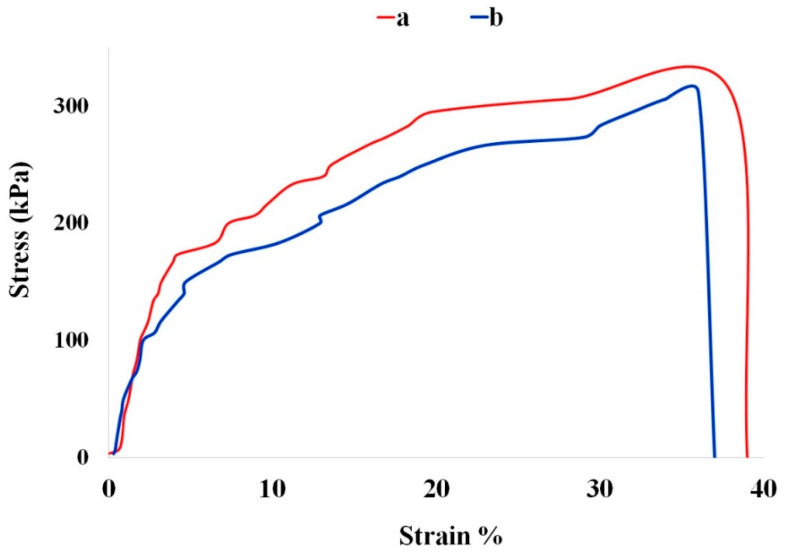
The stress–strain curves of CMC/Gel (**a**) and CMC/Gel/Cef (**b**) hydrogels.

**Figure 4 gels-11-00038-f004:**
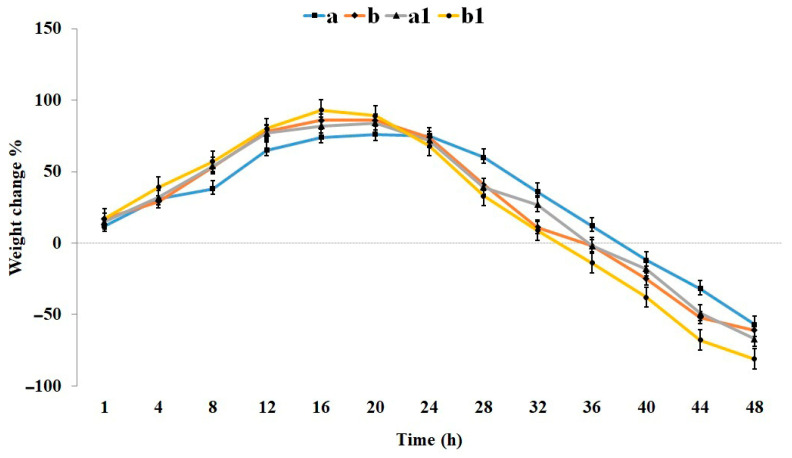
The percentage weight change in the hydrogels over 48 h at different pH levels: (**a**) CMC/Gel at pH 7.4, (**b**) CMC/Gel/Cef at pH 7.4, (**a1**) CMC/Gel at pH 9.0, and (**b1**) CMC/Gel/Cef at pH 9.0.

**Figure 5 gels-11-00038-f005:**
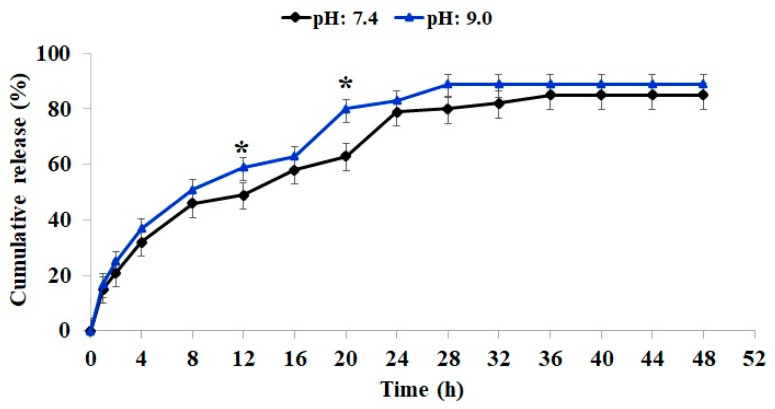
Cumulative release % of Cef from CMC/Gel hydrogel at different pHs. * *p* ˂ 0.05.

**Figure 6 gels-11-00038-f006:**
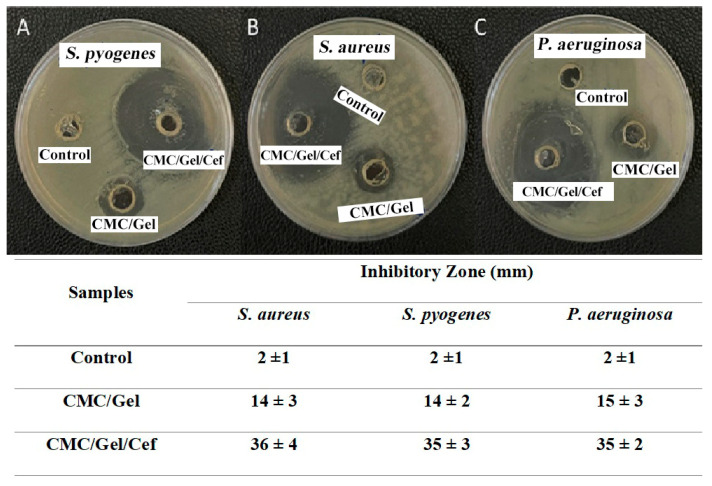
Inhibitory zone images and diameters of control and the hydrogels against *S. pyogenes* (**A**), *S. aureus* (**B**), and *P. aeruginosa* (**C**).

**Figure 7 gels-11-00038-f007:**
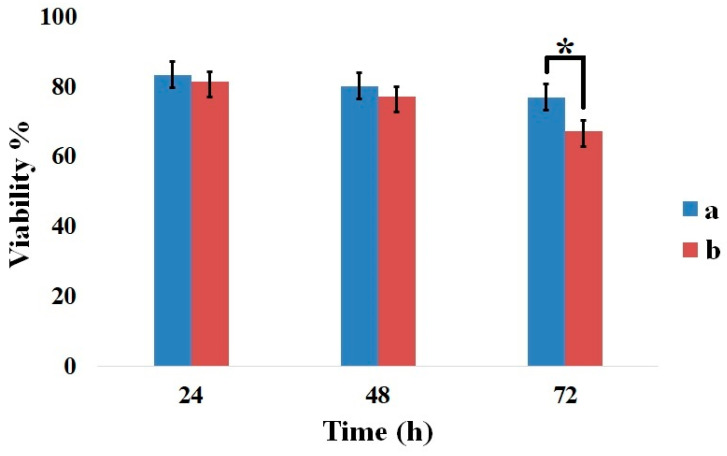
HFF viability % after different times. (**a**) CMC/Gel-treated and (**b**) CMC/Gel/Cef-treated groups, * *p* ˂ 0.05.

**Figure 8 gels-11-00038-f008:**
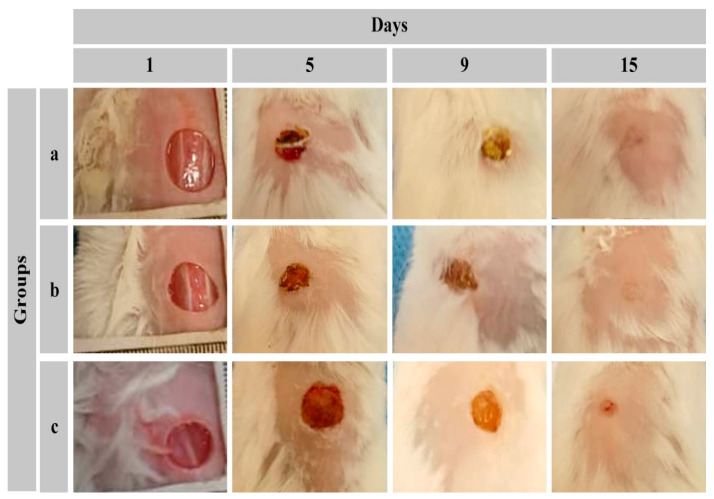
Wound area images were captured at different time points for (**a**) CMC/Gel-treated, (**b**) CMC/Gel/Cef-treated, and (**c**) control groups.

**Figure 9 gels-11-00038-f009:**
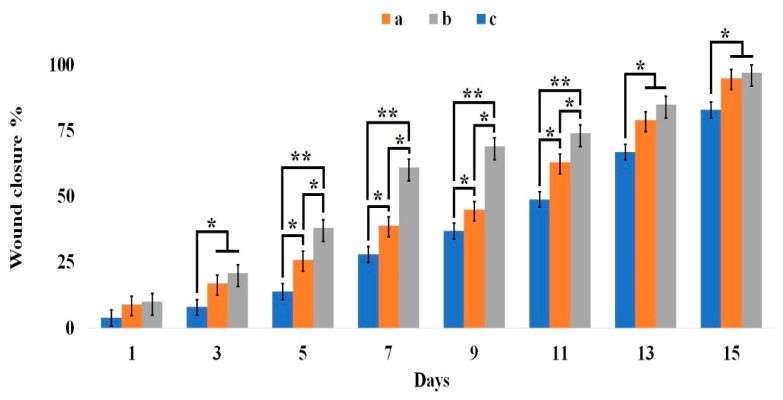
Wound closure percentage in mice was measured on different days for (**a**) CMC/Gel-treated, (**b**) CMC/Gel/Cef-treated, and (**c**) control groups. * *p* ˂ 0.05 and ** *p* ˂ 0.01.

**Figure 10 gels-11-00038-f010:**
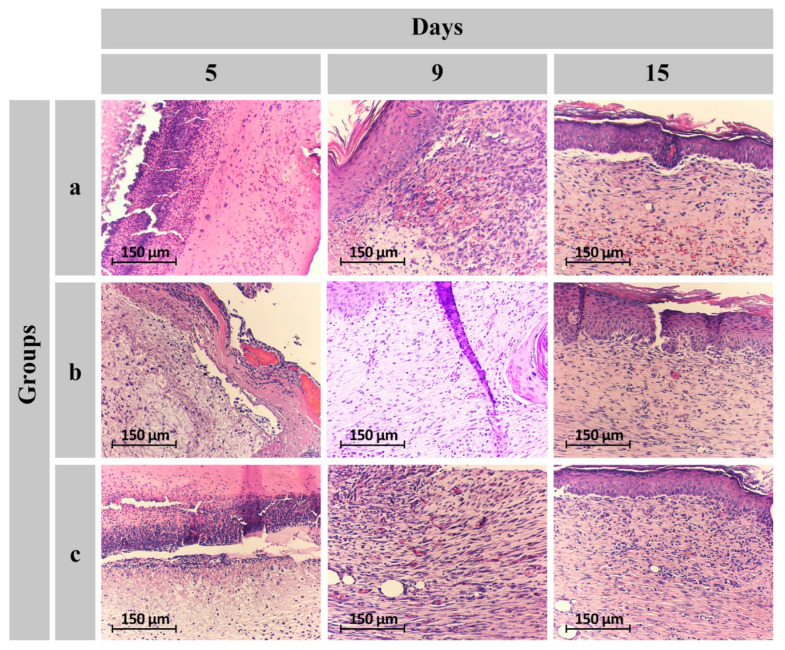
Representative H&E-stained histological sections at 100× magnification from postoperative days are shown for (**a**) CMC/Gel-treated, (**b**) CMC/Gel/Cef-treated, and (**c**) control groups.

**Table 1 gels-11-00038-t001:** Semi-quantitative histopathological analysis of healing wound sites was performed on different days for (a) CMC/Gel-treated, (b) CMC/Gel/Cef-treated, and (c) control groups.

Assayed Factors	5th Day			9th Day			15th Day		
	a	b	c	a	b	c	a	b	c
Epithelization	0	0	0	2	2	1	3	4	3
Inflammatory Cells	3	3	3	2	2	3	2	1	2
New Vessels	3	3	3	2	2	3	2	2	2
Fibroblasts	1	1	1	2	3	1	3	4	3
Collagen	0	1	0	2	2	1	3	3	2

## Data Availability

Data are available upon request.

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
