# Peer review of "Evaluation of Carboxymethyl Cellulose/Gelatin Hydrogel-Based Dressing Containing Cefdinir for Wound Healing Promotion in Animal Model"

_gels, 2025, doi:10.3390/gels11010038_

Round 1
Reviewer 1 Report
Comments and Suggestions for Authors
This study investigates the development of a hydrogel wound dressing composed of carboxymethyl cellulose and gelatin, incorporating cefdinir via a physical cross-linking method. The results demonstrate that the addition of Cef significantly boosts the antibacterial activity of the dressing, showing up to a 2.5-fold increase in efficacy against pathogens. The findings in this paper finally suggest that the CMC/Gel hydrogel dressing containing cefdinir is a biocompatible, effective option for wound healing, offering desirable physicochemical and antibacterial properties.
Overall, this study is of interesting. The CMC/Gel hydrogel is meaningful in the field of wound regeneration. I would like to support this manuscript if the authors can solve the following issues.
1. The authors need to test the drug loading rate and drug release rate for their CMC/Gel hydrogel.
2. In Figure 2, the authors need to use arrow to label the special FTIR peaks of Cef.
3. In Figure 10, the HE staining image of group B (9 days) is a little bit strange. The epidermis is peeled off. Please expain the reasons.
4. All of the statements in the introduction section should be supported by references. It is highly recommended to cite this reference (10.1002/adma.202400310) regarding the effective hydrogels in the sentence "Hydrogels are favored due to their biocompatibility, swelling properties, and mechanical similarity to extracellular matrices. They also excel in secretion absorption, oxygen transfer, and deformation, which are crucial for wound healing.".
Author Response
Reviewer 1
This study investigates the development of a hydrogel wound dressing composed of carboxymethyl cellulose and gelatin, incorporating cefdinir via a physical cross-linking method. The results demonstrate that the addition of Cef significantly boosts the antibacterial activity of the dressing, showing up to a 2.5-fold increase in efficacy against pathogens. The findings in this paper finally suggest that the CMC/Gel hydrogel dressing containing cefdinir is a biocompatible, effective option for wound healing, offering desirable physicochemical and antibacterial properties.
Overall, this study is of interesting. The CMC/Gel hydrogel is meaningful in the field of wound regeneration. I would like to support this manuscript if the authors can solve the following issues.
- The authors need to test the drug loading rate and drug release rate for their CMC/Gel hydrogel.
- We appreciate your valuable comment. The drug release rate was provided into the manuscript to comply with the comment (please see section 2.3.5 and 3.5).
- In Figure 2, the authors need to use arrow to label the special FTIR peaks of Cef.
- Thank you for the valuable comment. The figure has been revised to comply with the comment.
- In Figure 10, the HE staining image of group B (9 days) is a little bit strange. The epidermis is peeled off. Please explain the reasons.
- We appreciate the comment. The figure has been replaced with a suitable image to comply with the comment.
- All of the statements in the introduction section should be supported by references. It is highly recommended to cite this reference (10.1002/adma.202400310) regarding the effective hydrogels in the sentence "Hydrogels are favored due to their biocompatibility, swelling properties, and mechanical similarity to extracellular matrices. They also excel in secretion absorption, oxygen transfer, and deformation, which are crucial for wound healing.".
- Thanks for your attention in this regard. The introduction section was rechecked, and more references, including the suggested ones, were added to comply with the comments.

Reviewer 2 Report
Comments and Suggestions for Authors
The authors present the fabrication of a carboxymethyl cellulose/gelatin hydrogel-based dressing containing cefdinir for wound healing promotion. They have to an extent characterized the resulting hydrogel and have shown the effect using an in vitro agar diffusion test as well as a mice wound healing model.
While the content of the paper is interesting and the experimental results are clearly shown, several issues have to be addressed before it can be recommended for publication.
General comments
There are already several commercially available wound dressings based on CMC fibers and others. Please clearly state which advanatages this new approach would have.
Moreover, use of animal products such as gelatin in medicine is controversially discussed. Please reflect upon that.
Similarly, the use of topical antibiotics is obsolet in several countries due to the development of resistance. Hence, the incorporation of cefdinir does not seem the obvious choice and needs to be critically discussed.
Specific comments
In paragraph 3.1 it is stated that a high porosity of the product is ideal for facilitating cell infiltration. However, wouldn't this lead to ingrowth of cells into the dressing material in vivo, which subsequently would disturb healing due to tissue damage upon removal of the dressing.
Data showing the release profile of cefdinir from the hydrogels need to be added.
In paragraph 3.7 on the animal study, it is stated that cefdinir increased wound healing due to the antimicrobial effect. However, none of the wounds (especially the control group) do not show signs of wound infection. In general, wounds in rodents are not prone to infection and mostly heal by wound contraction. It is recommended to add a more detailed discussion on the subject. Moreover, it is stated in the second to last section of this paragraph that "the decline in angiogenesis" suggests "accelerated wound healing". It is necessary to add more detail and references here as well.
Author Response
Reviewer 2
The authors present the fabrication of a carboxymethyl cellulose/gelatin hydrogel-based dressing containing cefdinir for wound healing promotion. They have to an extent characterized the resulting hydrogel and have shown the effect using an in vitro agar diffusion test as well as a mice wound healing model.
While the content of the paper is interesting and the experimental results are clearly shown, several issues have to be addressed before it can be recommended for publication.
General comments
There are already several commercially available wound dressings based on CMC fibers and others. Please clearly state which advantages this new approach would have.
- We thank the reviewers for their valuable comments.
- Wound dressings fabricated from CMC fibers often exhibit inadequate mechanical properties for effective skin wound healing. To address this limitation, gelatin was incorporated into the formulation as a primary strategy to enhance mechanical strength. Additionally, gelatin is known to possess hemostatic properties and promote cellular proliferation, thereby accelerating wound healing. Furthermore, gelatin has been shown to improve the performance of wound dressings in chronic infections.
- In this study, we investigated the potential of gelatin-based CMC hydrogels as a delivery system for antibiotics to reduce the risk of infection. To enhance the biocompatibility of the hydrogel, a less cytotoxic crosslinking agent was employed as an alternative to glutaraldehyde. The resulting wound dressings are expected to offer superior performance in terms of mechanical properties, hemostatic efficacy, and antimicrobial activity.
Moreover, use of animal products such as gelatin in medicine is controversially discussed. Please reflect upon that.
- We appreciate your attention to this matter.
- Gelatin, a natural biopolymer derived from collagen, has been extensively utilized in biomedical applications due to its unique properties such as biocompatibility, hemostatic properties, and cell adhesion [1]. While concerns regarding its immunogenicity have been raised, particularly in long-term implants, its application in topical wound dressings, where it is intended for short-term use and rapid degradation, mitigates such risks [2]. In the context of the present wound dressing design, the limited exposure time and biodegradability of the gelatin-based hydrogel minimize the potential for adverse immune responses, making it a suitable choice for promoting wound healing.
- C Echave, Mari, et al. "Gelatin as biomaterial for tissue engineering." Current pharmaceutical design 23.24 (2017): 3567-3584.
- Krüger-Genge, A., et al. "Immunocompatibility and non-thrombogenicity of gelatin-based hydrogels." Clinical hemorheology and microcirculation 77.3 (2021): 335-350.
Similarly, the use of topical antibiotics is obsolete in several countries due to the development of resistance. Hence, the incorporation of cefdinir does not seem the obvious choice and needs to be critically discussed.
- The authors appreciate the reviewer's concern regarding the potential for antibiotic resistance. However, it is important to note that in certain clinical scenarios, topical antibiotics may be necessary as a last resort. The hydrogel formulation minimizes the risk of systemic antibiotic exposure by providing localized drug delivery. Additionally, the efficacy of the antibiotic-free hydrogel has been demonstrated in both the in vitro and in vivo studies, highlighting its potential for non-infected wound healing.
- In response to the reviewer's valuable feedback, a comprehensive discussion has been added to the Discussion section, addressing the concerns raised regarding antibiotic resistance and the role of the hydrogel in wound healing.
Specific comments
In paragraph 3.1 it is stated that a high porosity of the product is ideal for facilitating cell infiltration. However, wouldn't this lead to ingrowth of cells into the dressing material in vivo, which subsequently would disturb healing due to tissue damage upon removal of the dressing.
- We thank the reviewer for their insightful comment. Given the significant swelling and daily replacement of the hydrogel, the probability of fibroblast cell migration into the dressing is indeed low. As such, we have revised the discussion section to more accurately reflect this limitation.
Data showing the release profile of cefdinir from the hydrogels need to be added.
- Thanks for the comment. The release profile was added to the manuscript to comply with the comment (please see section 2.3.5 and 3.5).
In paragraph 3.7 on the animal study, it is stated that cefdinir increased wound healing due to the antimicrobial effect. However, none of the wounds (especially the control group) do not show signs of wound infection. In general, wounds in rodents are not prone to infection and mostly heal by wound contraction. It is recommended to add a more detailed discussion on the subject. Moreover, it is stated in the second to last section of this paragraph that "the decline in angiogenesis" suggests "accelerated wound healing". It is necessary to add more detail and references here as well.
- We thank the reviewer for their valuable comments. We have revised the discussion sections as suggested, providing relevant references to address the concerns.

Reviewer 3 Report
Comments and Suggestions for Authors
The study utilizes a specific ratio of carboxymethyl cellulose (CMC) and gelatin (Gel) for hydrogel preparation. However, it is not clear how the authors determined this particular ratio. Was it based on prior literature, preliminary optimization studies, or specific performance requirements? Providing a rationale for the chosen material ratio would help readers better understand the design process and its implications for the hydrogel's properties.
The swelling and degradation properties were evaluated but could benefit from additional pH-dependent studies, as wound environments vary.
There appears to be a discrepancy in the swelling and degradation data. The hydrogel is reported to not reach its equilibrium swelling weight within 24 hours, while simultaneously, approximately 80% of the hydrogel is degraded by the same time. This raises questions about the structural integrity of the hydrogel during degradation and its capacity to continue swelling. Such behavior seems counterintuitive, as significant degradation would typically compromise the hydrogel's ability to swell. This discrepancy should be clarified by either reconciling the experimental conditions for both tests, providing additional evidence (e.g., SEM images) of the hydrogel’s structure during the swelling and degradation phases, or discussing the potential reasons for this apparent contradiction. Additionally, simultaneous swelling and degradation studies under identical conditions could help resolve this issue and strengthen the interpretation of the data.
The disc diffusion method effectively demonstrates antibacterial activity, but additional methods (e.g., MIC or biofilm disruption studies) would strengthen the claims.
The manuscript provides valuable data on the mechanical properties of the hydrogels. However, it does not specify the number of specimens used for tensile testing. This information is critical to assess the reliability and statistical significance of the results. Could the authors clarify the number of specimens tested and whether replicates were performed to ensure reproducibility?
The SEM images provide important insights into the surface and cross-sectional morphology of the hydrogels. However, the scale bar in the SEM images appears to be too small, which may limit the ability to accurately interpret the structural details. Increasing the visibility and size of the scale bar or providing magnification details directly in the figure captions would enhance the clarity and utility of these images.
The inclusion of Cefdinir at 0.4% in the hydrogel composition raises questions about its potential impact on the swelling behavior. Cefdinir could influence the hydrogel's porosity, cross-linking density, or hydrophilicity, thereby altering its water absorption capacity. Additionally, the manuscript does not evaluate the release behavior of Cefdinir from the hydrogel, which is crucial for understanding its therapeutic effectiveness and impact on swelling over time. The authors are encouraged to investigate and report the Cefdinir release profile to provide a comprehensive understanding of its interaction with the hydrogel matrix and its potential effects on the material's swelling behavior.
Comments on the Quality of English Languagetypo-grammatical check
Author Response
Reviewer 3
The study utilizes a specific ratio of carboxymethyl cellulose (CMC) and gelatin (Gel) for hydrogel preparation. However, it is not clear how the authors determined this particular ratio. Was it based on prior literature, preliminary optimization studies, or specific performance requirements? Providing a rationale for the chosen material ratio would help readers better understand the design process and its implications for the hydrogel's properties.
- We thank the reviewer for their valuable comment. The formulation ratios were determined based on established literature in the field and further optimized through experimentation. Relevant references have been incorporated into the construction section to support these choices.
The swelling and degradation properties were evaluated but could benefit from additional pH-dependent studies, as wound environments vary.
- Thanks for the comment. Based on the acute and chronic wound pH values (i.e., 7.4-8.9), the swelling and degradation properties of the hydrogels were evaluated at various pH levels to address the reviewer's suggestion.
There appears to be a discrepancy in the swelling and degradation data. The hydrogel is reported to not reach its equilibrium swelling weight within 24 hours, while simultaneously, approximately 80% of the hydrogel is degraded by the same time. This raises questions about the structural integrity of the hydrogel during degradation and its capacity to continue swelling. Such behavior seems counterintuitive, as significant degradation would typically compromise the hydrogel's ability to swell. This discrepancy should be clarified by either reconciling the experimental conditions for both tests, providing additional evidence (e.g., SEM images) of the hydrogel’s structure during the swelling and degradation phases, or discussing the potential reasons for this apparent contradiction. Additionally, simultaneous swelling and degradation studies under identical conditions could help resolve this issue and strengthen the interpretation of the data.
- The authors thank the reviewer for their valuable comment. Swelling and degradation studies were conducted concurrently under identical conditions, and the manuscript has been revised accordingly.
The disc diffusion method effectively demonstrates antibacterial activity, but additional methods (e.g., MIC or biofilm disruption studies) would strengthen the claims.
- Thank you for your valuable feedback. Cefdinir is a well-established antibiotic with extensive research on its pharmacokinetic and pharmacodynamic properties. As this study primarily focuses on the synthesis and evaluation of the hydrogel, and the drug's structure remains unaltered, a detailed investigation into its antibacterial activity may not be essential within the scope of this work.
The manuscript provides valuable data on the mechanical properties of the hydrogels. However, it does not specify the number of specimens used for tensile testing. This information is critical to assess the reliability and statistical significance of the results.
Could the authors clarify the number of specimens tested and whether replicates were performed to ensure reproducibility?
- The authors thank the reviewer for their comments. All mechanical property measurements were performed in triplicate. The manuscript has been revised accordingly. Please see section 2.3.3 in the revised version.
The SEM images provide important insights into the surface and cross-sectional morphology of the hydrogels. However, the scale bar in the SEM images appears to be too small, which may limit the ability to accurately interpret the structural details. Increasing the visibility and size of the scale bar or providing magnification details directly in the figure captions would enhance the clarity and utility of these images.
- Thanks for the comment. The figure was amended to comply with the comment.
The inclusion of Cefdinir at 0.4% in the hydrogel composition raises questions about its potential impact on the swelling behavior. Cefdinir could influence the hydrogel's porosity, cross-linking density, or hydrophilicity, thereby altering its water absorption capacity. Additionally, the manuscript does not evaluate the release behavior of Cefdinir from the hydrogel, which is crucial for understanding its therapeutic effectiveness and impact on swelling over time. The authors are encouraged to investigate and report the Cefdinir release profile to provide a comprehensive understanding of its interaction with the hydrogel matrix and its potential effects on the material's swelling behavior.
- Thank you for your comment. We have repeated the swelling and degradation experiments for both hydrogels (with and without cefdinir) and revised the relevant section. Additionally, we have included the release profile of cefdinir in the manuscript (please see section 2.3.5 and 3.5).

Round 2
Reviewer 1 Report
Comments and Suggestions for Authors
All issues are solved. I would like to accept this manuscript.
Author Response
Thank you so much for your consideration.
Reviewer 2 Report
Comments and Suggestions for Authors
The authors have fully addressd my comments and revised their manuscript accordingly. It can now be recommended for publication.
Author Response
Thank you so much for your attention.
Reviewer 3 Report
Comments and Suggestions for Authors
The authors have addressed all comments, and I recommend publishing the article.
Comments on the Quality of English LanguageN/A
Author Response
Thank you so much for your attention.